# Nanoribbons self-assembled from short peptides demonstrate the formation of polar zippers between β-sheets

Meng Wang[1], Jiqian Wang[1], Peng Zhou[1], Jing Deng[2], Yurong Zhao[1], Yawei Sun[1], Wei Yang[1], Dong Wang[1], Zongyi Li[3], Xuzhi Hu[3], Stephen M. King[4], Sarah E. Rogers[4], Henry Cox[3], Thomas A. Waigh[3], Jun Yang[2], Jian Ren Lu[3] & Hai Xu[1]

Peptide self-assembly is a hierarchical process, often starting with the formation of α-helices, β-sheets or β-hairpins. However, how the secondary structures undergo further assembly to form higher-order architectures remains largely unexplored. The polar zipper originally proposed by Perutz is formed between neighboring β-strands of poly-glutamine via their side-chain hydrogen bonding and helps to stabilize the sheet. By rational design of short amphiphilic peptides and their self-assembly, here we demonstrate the formation of polar zippers between neighboring β-sheets rather than between β-strands within a sheet, which in turn intermesh the β-sheets into wide and flat ribbons. Such a super-secondary structural template based on well-defined hydrogen bonds could offer an agile route for the construction of distinctive nanostructures and nanomaterials beyond β-sheets.

[1] Centre for Bioengineering and Biotechnology, College of Chemical Engineering, China University of Petroleum (East China), 66 Changjiang West Road, Qingdao 266580, China. [2] National Center for Magnetic Resonance in Wuhan, Key Laboratory of Magnetic Resonance in Biological Systems, State Key Laboratory of Magnetic Resonance and Atomic and Molecular Physics, Wuhan Institute of Physics and Mathematics, Chinese Academy of Sciences, Wuhan 430071, China. [3] Biological Physics Group, School of Physics and Astronomy, The University of Manchester, Manchester M13 9PL, UK. [4] ISIS Pulsed Neutron Source, STFC Rutherford Appleton Laboratory, Didcot, Oxon OX11 0QX, UK. Correspondence and requests for materials should be addressed to J.W. (email: jqwang@upc.edu.cn) or to J.R.L. (email: j.lu@manchester.ac.uk) or to H.X. (email: xuh@upc.edu.cn)

Peptide and protein self-assembly provide a potential strategy for the in vitro fabrication of higher ordered structures and functional materials with biologically relevant characteristics (e.g., biocompatibility, biodegradability, biofunctionality) from the bottom-up. A considerable number of peptide-based materials tailored to specific applications, ranging from three-dimensional (3D) cell culture scaffolds to artificial enzymes, antibacterial and anticancer agents, hemostasis treatments, optical waveguides, and semiconductors, have been created through peptide self-assembly in the past few years[1–9]. Similar to protein folding into native architectures, peptide self-assembly is also a hierarchical process, typically starting with the formation of secondary structures such as α-helices, β-sheets or β-hairpins[10–13]. Although side-chain interactions are widely perceived to play a crucial role in the subsequent self-assembly events, how they dictate secondary structures to organize higher levels of scale and increased complexity so that various nanostructures and nano-materials can be fabricated in a controlled manner remains largely unexplored.

The short amphiphilic peptide Ac-I$_3$K-NH$_2$ self-assembles into flexible and twisted fibers with high stability, primarily due to the high hydrophobicity of isoleucine (Ile or I) and its strong propensity for β-sheet structuring[14,15]. Upon insertion of a glutamine (Gln or Q) residue at the interface between the hydrophobic I$_3$ motif and the hydrophilic lysine (Lys or K) residue, the resulting Ac-I$_3$QGK-NH$_2$ tends to form wide and flat ribbons[6]. Because both short peptides adopt β-sheet conformations, the dramatic variation in self-assembled nanostructures might imply different modes of interactions of the β-sheets.

Here, we design a series of peptides based on the general formula Ac-I$_3$XGK-NH$_2$, with X being either an uncharged polar amino acid (Q, serine: Ser or S, and asparagine: Asn or N) or a hydrophobic one (glycine: Gly or G, norvaline: Nva or $^{nor}$V, and leucine: Leu or L). We demonstrate that insertion of an uncharged polar residue at the intramolecular hydrophobic/hydrophilic interface can promote the formation of well-defined super-secondary structures based on side chain hydrogen bonding (H-bonding) interactions among β-sheets (defined as polar zippers), which eventually lead to distinctive nanoribbons, whilst insertion of hydrophobic amino acid residues results only in amyloid-like nanofibrils. This work thus demonstrates the capability and future potential of designing and fabricating distinctive nanostructures and nanomaterials by controlling structure and interaction processes beyond just β-sheets.

## Results

### From self-assembled nanofibers to nanoribbons.

Fig. 1 shows the molecular structure of the designed amphiphilic peptides. Transmission electron microscopy (TEM) imaging clearly reveals a ribbon-like morphology formed in the Ac-I$_3$QGK-NH$_2$ solution (8 mM, water, pH 7.0), and the width distribution histogram indicates a relatively uniform width around 50 nm, as shown in Fig. 2a. Except for a few kinked or tilted ribbons (red arrow in Fig. 2a), Ac-I$_3$QGK-NH$_2$ ribbons were generally observed to be rigid and flat, with little twists on them. Height profiles from atomic force microscopy (AFM) images further confirm the dominance of rigid and flat ribbons (Fig. 2b). The occurrence of the wide and flat plateau in the cross-sectional height profiles (the right panel of Fig. 2b) is consistent with their ribbon nature. Furthermore, the sectional analysis reveals heights of approximately 4, 8, and 12 nm, suggesting that the ribbons are likely multilayered. As shown in Fig. 2c, tapping-mode AFM height, amplitude, and phase images provide a thorough description of the multilayered feature of the Ac-I$_3$QGK-NH$_2$ ribbons. Terraces on a ribbon are clear in the amplitude and phase imaging. The

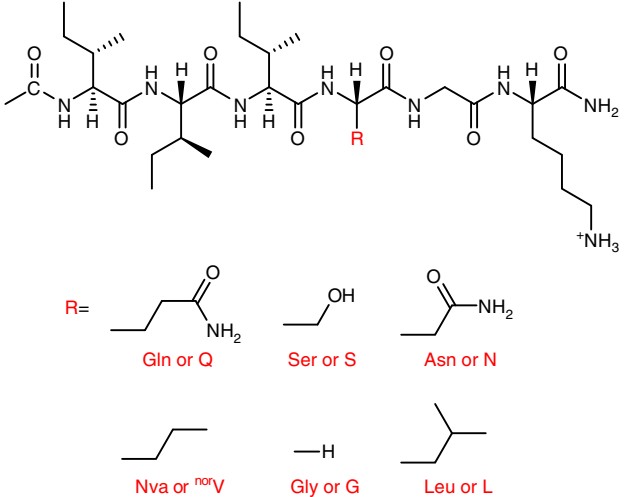

**Fig. 1** The general molecular structure of the designed short peptides Ac-I$_3$XGK-NH$_2$. Indicative side chain substitutions are marked by R. X is Q, S, N, $^{nor}$V, G, and L

terrace height changes can be well delineated using height profiling along the ribbon, giving rise to three terrace levels with a consistent step height of ~3.8 nm (the right panel of Fig. 2c). Such a step height variation is markedly larger than the extended length of Ac-I$_3$QGK-NH$_2$ (~2.45 nm), but less than twice the molecular length, implying that each terrace might be composed of an interdigitated peptide bilayer. Although there are usually errors associated with AFM height measurements, the bilayer thickness of Ac-I$_3$QGK-NH$_2$ was found to be mostly between 3.5 and 4.0 nm based on AFM sectional height profiling of ~30 individual ribbons (Supplementary Figure 1).

Small-angle neutron scattering (SANS) allows us to undertake in situ measurements of the self-assembled nanostructures in solution at length scales typically below 100 nm with statistical significance[16–21]. In SANS measurements, the intensity of neutron scattering, $I(q)$, is recorded as a function of the scattering vector modulus $q$ (Fig. 3), determined by the incident neutron wavelength $\lambda$ and the scattering angle $\theta$ by

$$q = \frac{4\pi}{\lambda}\sin\left(\frac{\theta}{2}\right) \tag{1}$$

For a dilute system of scattering nanoobjects where interactions between different objects are negligible, $I(q)$ can be described by

$$I(q) = N_p V_p^2 (\Delta\rho)^2 P(q) + I_b \tag{2}$$

where $N_p$ and $V_p$ are the number density and the volume of scattering objects, respectively (the volume fraction $\phi = N_p V_p$); $\Delta\rho$ denotes the difference in neutron scattering length density between the scattering object and the solvent matrix; $P(q)$, the form factor, describes the size and shape of scattering objects; and $I_b$ is the residual $q$-independent background intensity.

For nanoribbons, an elliptical cylinder model (ECM) was applied to fit the scattering data using the SasView program (http://www.sasview.org). As exemplified by the magenta dashed line in Fig. 3a, an elliptical cylinder gave an approximate description of the scattering data from 8 mM Ac-I$_3$QGK-NH$_2$ at $q$ values of less than 0.07 Å$^{-1}$ and larger than 0.2 Å$^{-1}$. In spite of a poor fit at the intermediate $q$ range from 0.07 to 0.2 Å$^{-1}$, such a model fit could approximate the main structural parameters of Ac-I$_3$QGK-NH$_2$ nanoribbons, giving a minor radius of 4.2 nm, an axial ratio of 4.8, and a length of more than 100 nm (Table 1 and

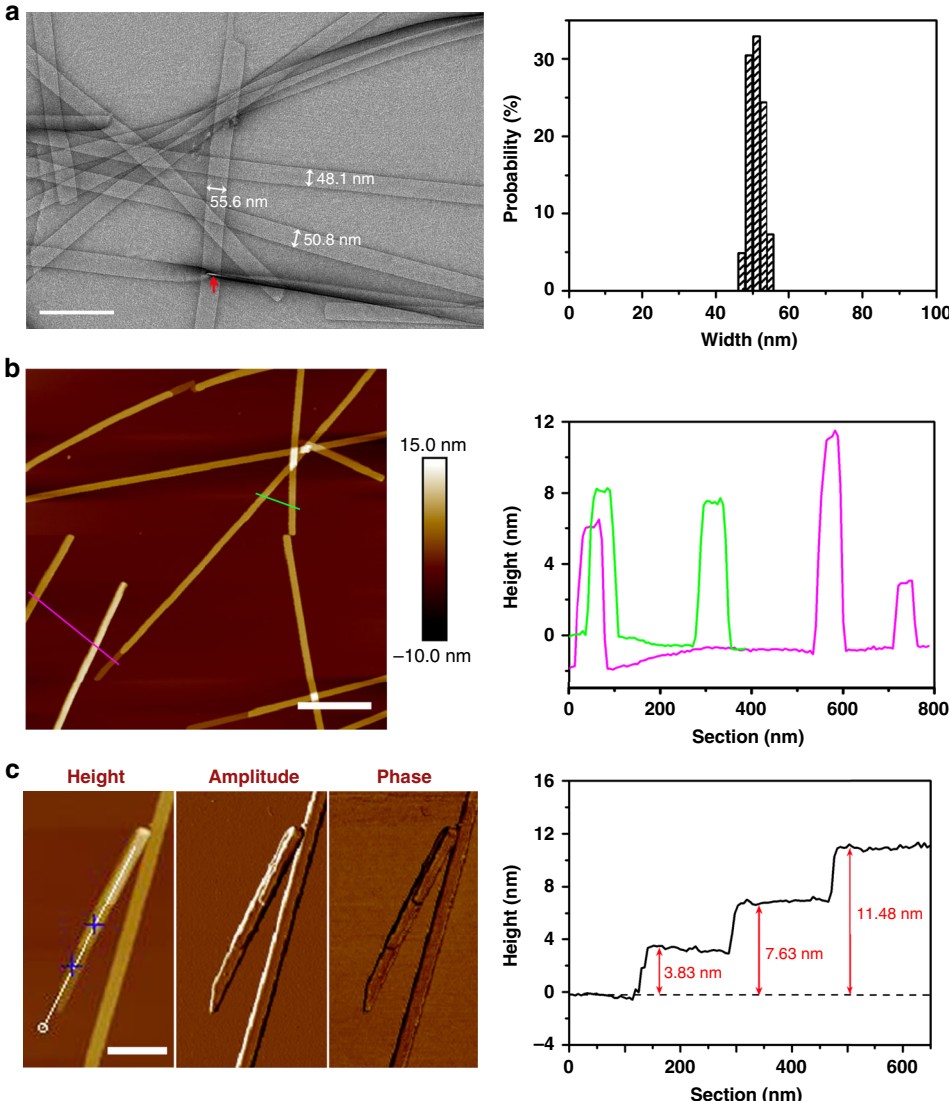

**Fig. 2** Nanoribbons formed by 8 mM Ac-I₃QGK-NH₂ at pH 7.0 after incubation for 1 week. **a** TEM micrograph (left panel) and width distribution histogram (right panel, based on measurements of ~100 individual ribbons). Scale bar, 200 nm. **b** Tapping-mode height AFM image (left panel) and representative cross-sectional height profiles (right panel). Scale bar, 500 nm. **c** Higher magnification tapping-mode AFM images (left panel: height, amplitude, and phase) and a height profile along the ribbon axis (right panel). Scale bar, 200 nm

Supplementary Table 1). Note that the model fitting was insensitive to dimensions of more than 100 nm because the scattering data did not extend to sufficiently low values of $q$[20,21]. The combination of two elliptical cylinders (ECM1 + ECM2) with different parameters was found to significantly improve the fitting quality (denoted as the blue line in Fig. 3a). As a result, two types of nanostructures could be revealed, one being long and wide nanoribbons with a minor radius of 4.3 nm, an axial ratio of 5.0, and a length of more than 100 nm (p1_ entries in Table 1 and Supplementary Table 1) and the other corresponding to very small cylindrical aggregates with a radius of 1.2 nm (axial ratio close to 1) and a length of 2.5 nm (p2_ entries in Table 1 and Supplementary Table 1). Furthermore, the agreement between the calculated and experimental intensities could be further improved when we incorporated a size distribution (polydispersity) during the fitting process, as demonstrated by the markedly reduced $\chi^2/N_{pts}$ values when $\sigma$/ are greater than zero (Table 1 and Supplementary Table 1; where $\sigma$ denotes the standard deviation of a lognormal distribution and < > denotes the mean

value). Because the fitting process was significantly complicated by the additional degrees of freedom, size polydispersity was only incorporated into the minor radius and axial ratio of the long and wide ribbons, giving an $\sigma$/<radius> of 0.15 and an $\sigma$/< ratio> of 0.22. As indicated above, the structural parameters derived from the SANS data are the averaged values (over several mm³ of solution) and have statistical significance. For the long and wide nanoribbons, the average thickness of 8.6 nm (2 × minor radius) and width of 43 nm (2 × minor radius × axial ratio) from the SANS data were broadly consistent with those from TEM and AFM measurements, all indicating the occurrence of multilayered stacking. The small aggregates are assumed to be the oligomeric peptide nanoobjects, most likely acting as the precursor of the large nanoribbons during self-assembly. However, they were too small to be observed in TEM and AFM imaging.

When the sample was diluted 4-fold, SANS measurements showed that the scattering signal decreased in intensity but changed little in shape (Fig. 3a). A single elliptical cylinder was found to be adequate to fit these data, giving rise to a minor

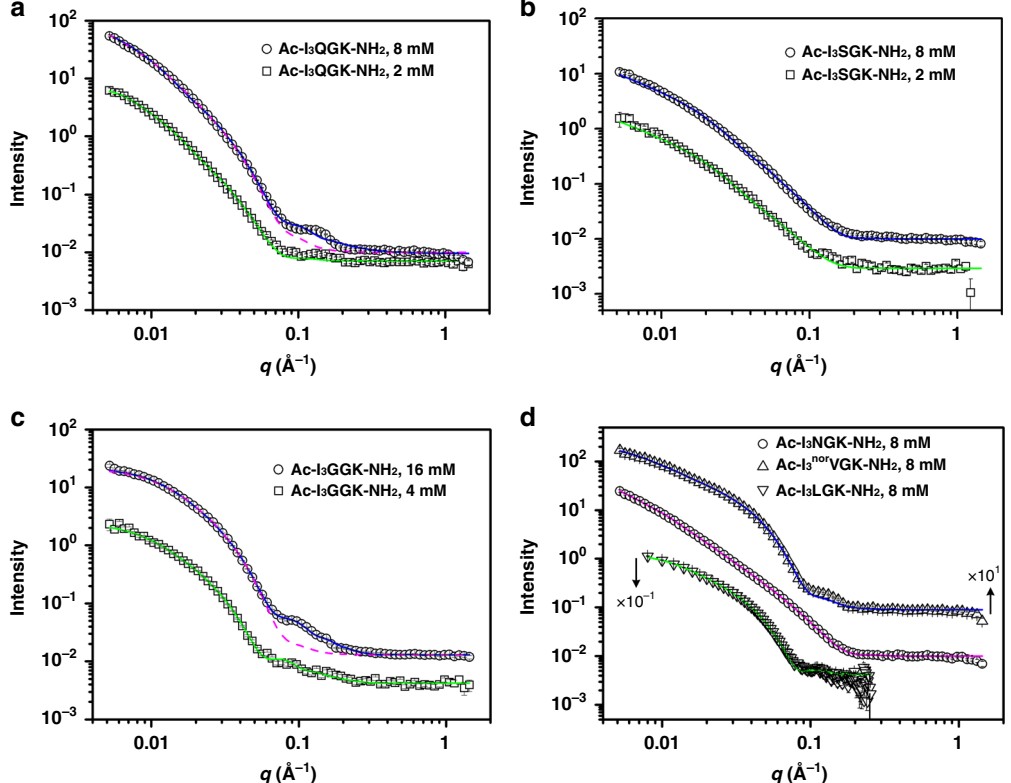

**Fig. 3** SANS data and fitted profiles. **a** Ac-I$_3$QGK-NH$_2$. **b** Ac-I$_3$SGK-NH$_2$. **c** Ac-I$_3$GGK-NH$_2$. **d** Ac-I$_3$NGK-NH$_2$, Ac-I$_3$$^{nor}$VG-NH$_2$, and Ac-I$_3$LGK-NH$_2$. The error bars are also shown, but in most cases, they are hidden within the symbols. For clarity, the signals of Ac-I$_3$$^{nor}$VG-NH$_2$ in (**d**) have been vertically displaced by a factor of $\times 10^1$, whereas those of Ac-I$_3$LGK-NH$_2$ by a factor of $\times 10^{-1}$. The SANS data of Ac-I$_3$LGK-NH$_2$ were recorded on the LOQ diffractometer which has a narrower $q$ range while those for the other peptides were obtained on the SANS2D diffractometer. The lines are the best fits to the models with parameters given in Table 1, and Supplementary Tables 1 and 2. Note that for 8 mM Ac-I$_3$QGK-NH$_2$ and 16 or 4 mM Ac-I$_3$GGK-NH$_2$, the combination of two ECM models or two FCM models, respectively, can fit the SANS data much better (solid lines). The dashed lines represent the best fits to a single ECM model and a single FCM model for 8 mM Ac-I$_3$QGK-NH$_2$ and 16 mM Ac-I$_3$GGK-NH$_2$, respectively

**Table 1 The fitting models applied to, and the optimal main structural parameters extracted from, the SANS data from Ac-I$_3$XGK-NH$_2$ (X=Q, S, and N) as shown in Fig. 3**

| Peptide & concentration | Ac-I$_3$QGK-NH$_2$ | | | Ac-I$_3$SGK-NH$_2$ | | Ac-I$_3$NGK-NH$_2$ |
|---|---|---|---|---|---|---|
| | 8 mM | 8 mM | 2 mM | 8 mM | 2 mM | 8 mM |
| Fitting model | ECM [a] | ECM + ECM | ECM | ECM | ECM | ECM |
| p1_Minor Radius (Å) | 42 | 43 | 42 | 16.5 | 15.4 | 16 |
| $\sigma$/<radius>[b] | 0.16 | 0.15 | 0.12 | 0.03 | 0.05 | 0.08 |
| p1_Axial ratio | 4.8 | 5 | 4.8 | 7 | 6.7 | 11.5 |
| $\sigma$/< ratio> | 0.16 | 0.22 | 0.17 | 0.32 | 0.34 | 0.39 |
| p1_Length (Å) | >1000 | >1000 | >1000 | >1000 | >1000 | >1000 |
| p2_Minor radius (Å) | / | 12 | / | / | / | / |
| p2_Axial ratio | / | 1.0 | / | / | / | / |
| p2_Length (Å) | / | 25 | / | / | / | / |
| $\chi^2$/N$_{pts}$[c] | 78.5 | 28.7 | 5.0 | 18.2 | 3.3 | 2.9 |

For clarity, other structural parameters are given in Supplementary Table 1. During the fitting process, the incorporation of size polydispersity into the minor radius and axial ratio of long and wide nanoribbons could improve the fitting quality but had little impact on their main structural parameters
[a]ECM denotes the elliptical cylinder model
[b]$\sigma$ = standard deviation of the lognormal distribution and < > = mean value
[c]$\chi^2$/N$_{pts}$ values were significantly decreased in the presence of size polydispersity for the minor radius and axial ratio of long and wide ribbons, in comparison with those assuming an absence of size polydispersity (Supplementary Table 1)

radius of 4.2 nm ($\sigma$/<radius > = 0.12), an axial ratio of 4.8 ($\sigma$/<ratio> = 0.17), and a length of more than 100 nm (denoted as the green line in Fig. 3a with the key parameters given in Table 1), suggesting that the main structural features of Ac-I$_3$QGK-NH$_2$ nanoribbons were well preserved after dilution. Furthermore, the

effective volume fraction was found to decrease, in rough proportion to the dilution (Supplementary Table 1). The preservation of the long and wide Ac-I$_3$QGK-NH$_2$ nanoribbons in the diluted solution was further confirmed from TEM imaging (Supplementary Figure 2a). The small aggregates (peptide

oligomers) did not make enough contribution to the scattering, or underwent disassembly upon dilution, so as to make them undetectable.

Ser is also a polar uncharged amino acid, with its side chain containing a β-hydroxyl group and being relatively small in size (Fig. 1). Upon insertion of a Ser residue, the resulting Ac-I$_3$SGK-NH$_2$ also assembled into long and rigid nanoribbons at 8 mM (Fig. 4). It is interesting that these Ac-I$_3$SGK-NH$_2$ nanoribbons were narrower than the Ac-I$_3$QGK-NH$_2$ ones, their width derived from TEM measurements being about 29 nm (Fig. 4a). However, the cross-sectional profile derived from the height AFM imaging showed a uniform height of ~3.8 nm (Fig. 4b), a value expected from the single interdigitated bilayer. Importantly, a single ECM fitted the corresponding SANS data well, giving an average ribbon thickness of 3.3 nm and an average width of 23 nm (denoted as the blue line in Fig. 3b and the key parameters given in Table 1), again broadly consistent with the structural parameters derived from TEM and AFM measurements. A further SANS measurement on the diluted Ac-I$_3$SGK-NH$_2$ solution (2 mM) showed that the scattering data remained unchanged in shape and the level of reduction was also consistent with the dilution. It also fitted to the same ECM (denoted as the green line in Fig. 3b), but with a slightly decreased ribbon thickness and width (Table 1). The presence of long Ac-I$_3$SGK-NH$_2$ nanoribbons in the diluted solution was further verified by TEM imaging (Supplementary Figure 2b).

Asn has one less carbon than Gln, but the same polar amide group in the side chain (Fig. 1). Ac-I$_3$NGK-NH$_2$ also assembled into long nanoribbons (Supplementary Figure 3a). The SANS data was again fitted to a long elliptical cylinder, giving an average ribbon thickness of 3.2 nm and an average width of 37 nm (denoted as the magenta line in Fig. 3d and the key parameters given in Table 1). Thus, the Ac-I$_3$NGK-NH$_2$ nanoribbons were narrower than the Ac-I$_3$QGK-NH$_2$ ones, but wider than the Ac-I$_3$SGK-NH$_2$ ones. Their thickness indicates that the Ac-I$_3$NGK-NH$_2$ nanoribbons also predominantly exist as single interdigitated bilayers, the same as the Ac-I$_3$SGK-NH$_2$ nanoribbons. Again, the incorporation of size polydispersity into the minor radius and axial ratio improved the fitting quality of the Ac-I$_3$NGK-NH$_2$ and Ac-I$_3$SGK-NH$_2$ nanoribbons (Table 1 and Supplementary Table 1).

**Back to nanofibers.** Gly has a single hydrogen atom in its side-chain with no obvious hydrophilic or hydrophobic property (Fig. 1). As a result, it has great conformational freedom and cannot interact significantly with other amino acids through side-chain interactions. The amino acid has been widely used as a spacer between functional motifs in peptide design. When Gly was substituted for the above polar residues (Gln, Ser, and Asn), the resulting Ac-I$_3$GGK-NH$_2$ self-assembled into nanofibers. TEM imaging revealed that the Ac-I$_3$GGK-NH$_2$ nanofibers were long and thin, with a width of approximately 12 nm (Fig. 5a). AFM imaging revealed that the nanofibers twisted in a left-handed manner (Fig. 5b and Supplementary Figure 4). These structural features of the Ac-I$_3$GGK-NH$_2$ nanofibers are consistent with those of Ac-I$_3$K-NH$_2$ nanofibers[15], suggesting that glycine insertion has had little effect. Furthermore, magnified AFM imaging and height profiling along the fiber axis revealed three types of left-handed nanofibers with different heights and pitches, as shown in Fig. 5b. This is expected since structural polymorphism has been extensively observed in amyloid fibrils[22–25]. Thicker fibers usually form via the association or stacking of thinner ones.

The Ac-I$_3$GGK-NH$_2$ solution used in the TEM and AFM measurements (Fig. 5a, b) was at a concentration of 16 mM instead of 8 mM. This was due to the weaker self-assembling ability of Ac-I$_3$GGK-NH$_2$ than the other peptides studied, as demonstrated from the SANS work. As shown in Fig. 3, the SANS

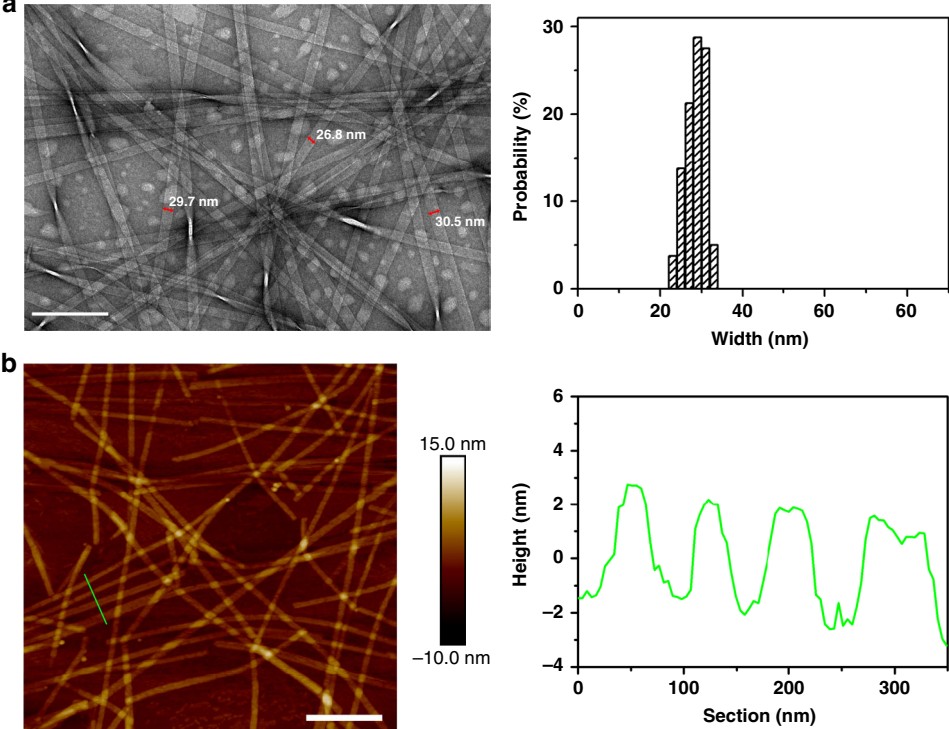

**Fig. 4** Nanoribbons formed by 8 mM Ac-I$_3$SGK-NH$_2$ at pH 7.0 after incubation for 1 week. **a** TEM micrograph (left panel) and width distribution histogram (right panel, based on measurements of ~100 nanoribbons). Scale bar, 200 nm. **b** Tapping-mode height AFM image (left panel) and representative cross-sectional profile (right panel). Scale bar, 500 nm

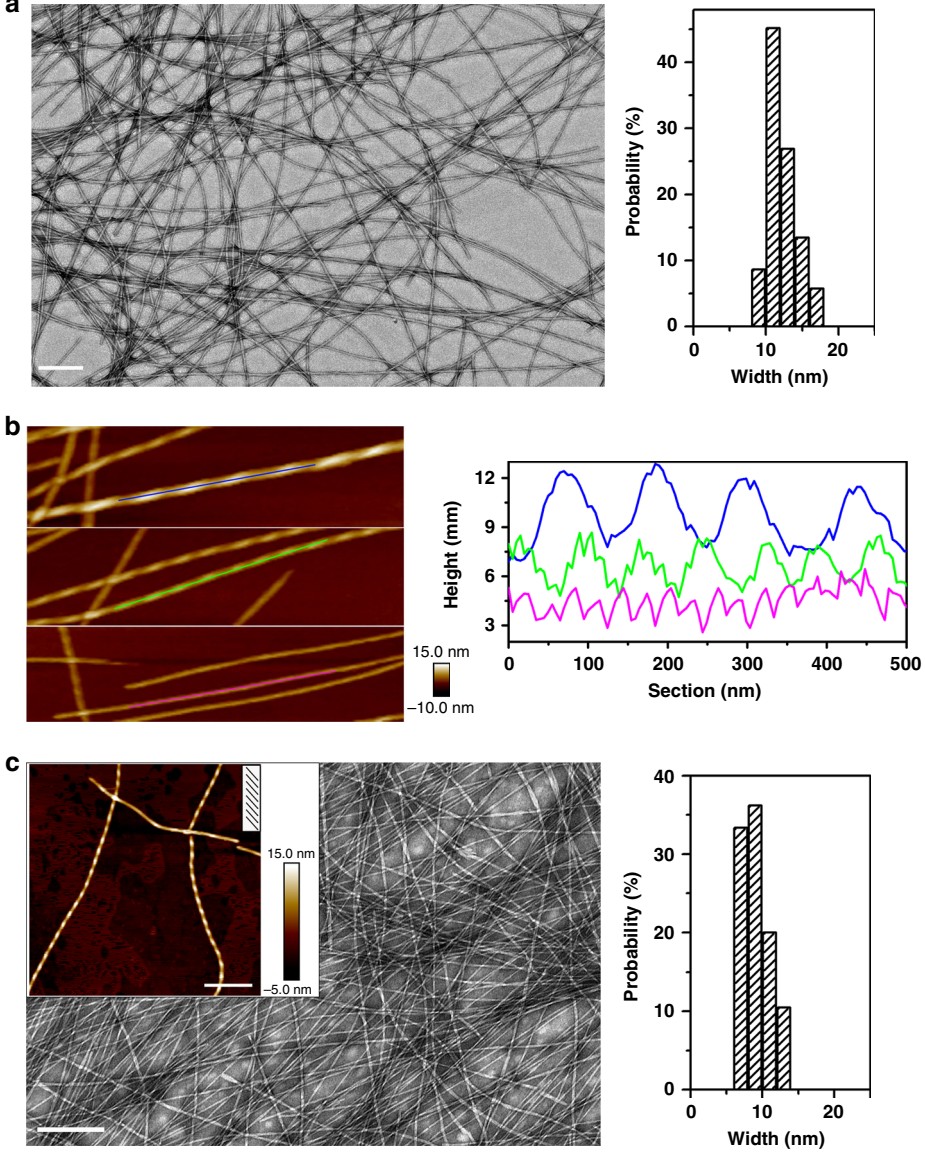

**Fig. 5** Nanofibers formed by Ac-I$_3$GGK-NH$_2$ and Ac-I$_3^{nor}$VGK-NH$_2$ at pH 7.0 after incubation for 1 week. **a** TEM micrograph (left panel) and width distribution histogram of 16 mM Ac-I$_3$GGK-NH$_2$ nanofibers (right panel, based on measurements of ~100 individual fibers). Scale bar, 200 nm. **b** Magnified height AFM image of Ac-I$_3$GGK-NH$_2$ nanofibers (left panel) and corresponding sectional height profiles along the fiber axis showing fiber polymorphism (right panel). **c** TEM micrograph (left panel) and width distribution histogram of 8 mM Ac-I$_3^{nor}$VGK-NH$_2$ nanofibers (right panel, based on measurements of ~100 nanoobjects), with an inset AFM imaging in the left panel indicating their left-handed twisting. Scale bar represents 200 and 300 nm the TEM and AFM images, respectively

intensity of Ac-I$_3$GGK-NH$_2$ at 16 mM was close to that of other peptides at 8 mM. A flexible cylinder model (FCM) was applied to fit the SANS data from the nanofibers. Just like Ac-I$_3$QGK-NH$_2$, a single FCM could not fit the scattering data perfectly, even in the presence of size polydispersity for the radius and Kuhn length (denoted by the magenta dash line in Fig. 3c with structural parameters given in Supplementary Table 2). In comparison, the combination of two FCMs (FCM1 + FCM2) was found to fit the scattering data well (denoted by the blue solid line in Fig. 3c with structural parameters shown in Supplementary Table 2). Data analysis helped to identify both long fibers with an averaged diameter of 10.4 nm ($\sigma/<$ radius$> = 0.08$) and lengths of more than 100 nm, and much thinner and shorter fibrils with a diameter of ~2.4 nm and length of ~36 nm. The long Ac-I$_3$GGK-NH$_2$ fibers were found to have a Kuhn length of 3.5 nm ($\sigma/<$Kuhn

length $> = 0.31$). The smaller peptide aggregates are likely protofibrils acting as intermediate filaments in templating long fibers[22–26]. The protofibrils could also be observed in AFM imaging (as indicated by green arrows in Supplementary Figure 5).

For the diluted Ac-I$_3$GGK-NH$_2$ solution at 4 mM, the fiber morphology and polymorphism were well preserved (Supplementary Figure 2c). Data analysis of the measured SANS data also indicated the coexistence of longer and thicker fibers with shorter and thinner ones (denoted as the green line in Fig. 3c with structural parameters given in Supplementary Table 2). The diameter of the thick fibers was slightly increased to 12.6 nm ($\sigma/<$radius$> = 0.07$), accompanied with an increase in its scattering length density ($\rho$) from $4.0 \times 10^{-6}$ to $4.5 \times 10^{-6}$ Å$^{-2}$ (Supplementary Table 2). The increase in $\rho$ of the Ac-I$_3$GGK-NH$_2$ fibers

suggests that more water ($D_2O$) molecules entered the nanofibers upon dilution, leading to their swelling and an increase in their diameters.

Nva is an isomer of valine, but its side-chain is unbranched. As a result, it exhibits a structural similarity to Gln, except for the additional amide group in the latter (Fig. 1). As shown in Fig. 5c, Ac-$I_3$$^{nor}$VGK-$NH_2$ self-assembled into long and thin nanofibers rather than wide nanoribbons. TEM imaging indicated a width of approximately 9 nm and AFM imaging showed that the Ac-$I_3$$^{nor}$VGK-$NH_2$ fibers had a left-handed twist. SANS data analysis revealed that a single FCM with size polydispersity (for the radius and Kuhn length) could fit the measured SANS data well, giving it an average diameter of 7.9 nm ($\sigma$/<radius> = 0.20) and a Kuhn length of 49.8 nm ($\sigma$/<Kuhn length> = 0.48) (denoted as the blue line in Fig. 3d with structural parameters given in Supplementary Table 2).

Leu is a $\gamma$-branched amino acid with two methyl groups on the $\gamma$-carbon. Its side-chain bears a structural geometry similar to that of Asn (Fig. 1), in spite of their completely different chemical properties. TEM imaging indicated the formation of long nanofibers from Ac-$I_3$LGK-$NH_2$ (Supplementary Figure 6a). SANS data analysis revealed that the nanofibers had an average diameter of 8.4 nm ($\sigma$/< radius> = 0.11) and a Kuhn length of 28.2 nm ($\sigma$/<Kuhn length> = 0.14) (denoted as the green line in Fig. 3d with structural parameters given in Supplementary Table 2).

The above peptide nanoribbons and nanofibers were typically formed within one week of incubation. There was little variation in their morphologies and dimensions with a further increase of incubation time, which could be demonstrated by AFM and SANS measurements (Supplementary Figure 7). Furthermore, the peptide nanostructures showed high stability in water and no precipitates or turbidity occurred even after three months of storage, in sharp contrast to amyloid fibrils.

**Secondary structures and molecular packing modes**. As demonstrated, the designed peptides (Ac-$I_3$XGK-$NH_2$) all self-assembled into one-dimensional (1D) nanostructures. Although these 1D nanostructures showed different morphologies and sizes with varying X residues, circular dichroism (CD) and FTIR measurements confirmed that all the peptides adopted $\beta$-sheet conformations. As shown in Fig. 6a, b and Supplementary Figures 3b and 6b, a positive CD maximum at ~200 nm, a negative CD minimum at ~220 nm, and an FTIR band at ~1620 $cm^{-1}$, characteristic of $\beta$-sheet secondary structures, are observed. In fact, it is mainly $\beta$-sheet H-bonding between peptide strands that drives the axial aggregation of peptide molecules (denoted as the $y$-direction) during the self-assembly of $\beta$-sheet-based 1D nanostructures[11,14,27].

Further powder X-ray diffraction (XRD) measurements indicated that these 1D peptide assemblies had a similar XRD pattern, i.e. two marked reflections at ~8.4° and ~19.0° (Supplementary Figure 8), corresponding to $d$-spacings of 10.5 and 4.7 Å, respectively. The 4.7 Å reflections are ascribed to the strand-strand separation in a $\beta$-sheet while the 10.5 Å reflections arise from the sheet-sheet separation (lamination distance) upon lateral stacking of $\beta$-sheets[28–31]. Clearly, the 1D nanostructures have a similar secondary structure to amyloid fibrils. The slight increase in the lamination distance from 9.9 Å for amyloid fibrils to 10.5 Å in this case might be caused by the bulky and abundant Ile side chains.

To determine peptide register within a $\beta$-sheet, we incorporated one 1-$^{13}$C- and one $^{15}$N-labeled Ile amino acid into the sequence of $I_3$QGK, giving rise to an isotope-labeled peptide [$^{15}$N]Ile1[1-$^{13}$C]Ile3-Ac-$I_3$QGK-$NH_2$. Given the anti-parallel

$\beta$-sheet arrangement, such a labeling protocol was selected to allow the determination of the interatomic distance between neighboring $\beta$-strands within a sheet, by using $^{13}$C{$^{15}$N} rotational-echo double-resonance (REDOR) NMR[28,32]. The inter-strand spacing between neighboring $\beta$-sheets was determined to be ~10.5 Å from XRD (Supplementary Figure 8) and such a distance had little impact on the inter-strand $^{13}$C-$^{15}$N distance measurements within a $\beta$-sheet in REDOR NMR[28]. Fig. 6c shows the $^1$H-$^{13}$C cross polarization (CP) spectrum of [$^{15}$N]Ile1[1-$^{13}$C]Ile3-Ac-$I_3$QGK-$NH_2$ nanoribbons collected from an 8 mM solution. The line width of ~120 Hz for $^{13}$C signals indicates a conformational homogeneity of the peptide assemblies. Fig. 6d shows experimental $^{13}$C{$^{15}$N} REDOR data and fitting curves. Molecular dynamics (MD) simulations indicated two inter-strand $^{13}$C-$^{15}$N distances of 7.4 and 8.1 Å for the one-residue shift packing mode (inset of Fig. 6d). The experimental REDOR data is in excellent agreement with the computed REDOR curve using the $^{13}$C-$^{15}$N distances from this packing mode with one-residue shift. When we varied the intermolecular distances by more than 0.3 Å (7.7/8.4 Å and 7.1/8.8 Å), the calculated REDOR curves (dashed lines) deviated markedly from the experimental data. On the other hand, MD simulations indicated two inter-strand $^{13}$C-$^{15}$N distances of 5.5 and 5.8 Å for the two-residue shift packing mode and of 4.2 and 5.6 Å for the three-residue shift packing mode (Supplementary Figure 9a). Based on these distances, the calculated REDOR curves differ hugely from the experimental data in shape and $S/S_0$ values, also shown in Supplementary Figure 9b. These results clearly indicate a one-residue shift packing mode for anti-parallel Ac-$I_3$QGK-$NH_2$ $\beta$-sheets.

Due to their different morphologies, these 1D nanostructures might have distinctive properties and potentials in technological applications, such as scaffolding and nanofabrication[11]. Given that charged Lys residues are mainly projected on their surfaces, our preliminary experiments have revealed, for example, that the Ac-$I_3$$^{nor}$VGK-$NH_2$ nanofibers readily template the synthesis of thin silica nanotubes, while the Ac-$I_3$QGK-$NH_2$ nanoribbons induced the formation of silica nanotapes under ambient aqueous conditions, as shown in Supplementary Figure 10.

**Discussion**

We demonstrate that the designed peptides self-assembled into 1D nanostructures with typical $\beta$-sheet conformations. However, marked differences in morphology and size suggest different packing modes and different degrees of $\beta$-sheet packing. In fact, the lateral stacking degree of $\beta$-sheets is dependent on the magnitude of the adhesive forces among amino acid side-chains, which compensate for the elastic energy required for untwisting them from their natural twisted states[10,11]. In the case of Ac-$I_3$K-$NH_2$, the lateral stacking of $\beta$-sheets is primarily driven by the close contact of Ile side-chains. Given that such a hydrophobic interaction is also responsible for offsetting the electrostatic repulsion among Lys side chains, it is not strong enough to significantly untwist the Ac-$I_3$K-$NH_2$ sheets, leading to their limited stacking and eventually the formation of thin nanofibers with remarkable twisting[15]. Upon insertion of an X residue between the hydrophobic $I_3$ motif and the charged Lys residue, the 1D nanostructures formed by Ac-$I_3$XGK-$NH_2$ peptides showed different morphologies and sizes, indicating that interfacial X plays a pivotal role in modulating $\beta$-sheet lateral stacking along the $x$-direction.

The designed peptides are structurally akin to common surfactants, and as a result, these amphiphilic molecules tend to adopt an anti-parallel packing mode upon associating into a $\beta$-sheet, thereby alleviating the electrostatic repulsion between

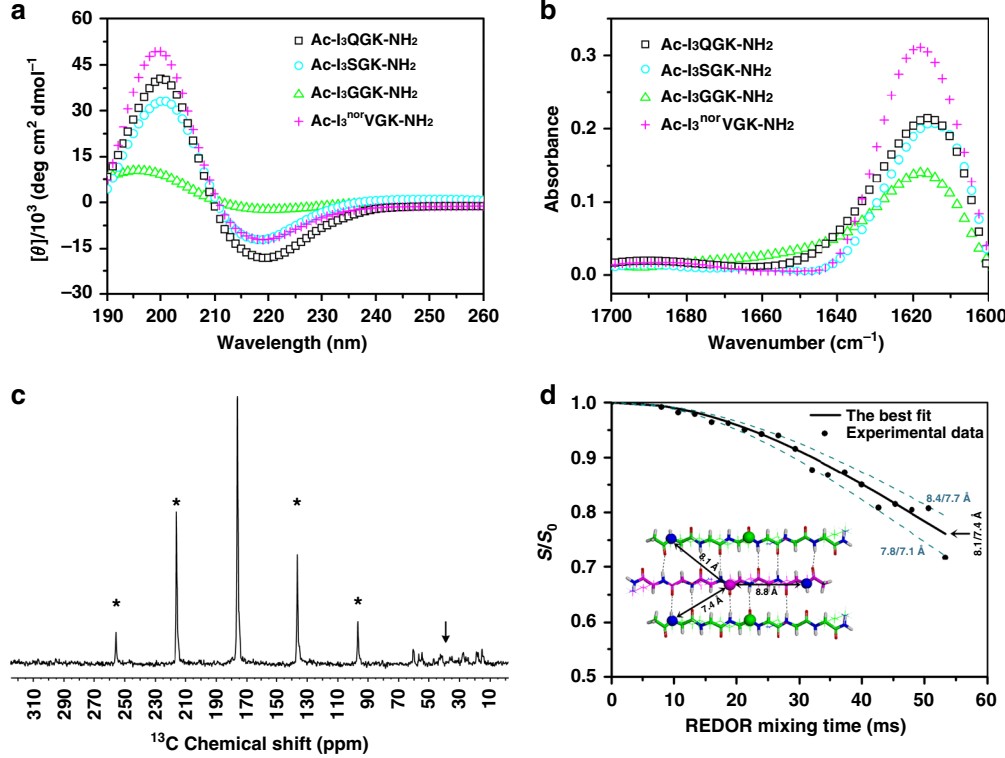

**Fig. 6** Secondary structures and molecular packing modes. **a, b** CD and FTIR spectra of the designed peptides at 8 mM except for Ac-I$_3$GGK-NH$_2$ at 16 mM. For clarity, the CD spectra of Ac-I$_3$NGK-NH$_2$ and Ac-I$_3$LGK-NH$_2$ are given in Supplementary Figures 3b and 6b, respectively, but also indicating the formation of β-sheet conformations. **c** $^1$H-$^{13}$C CP spectrum of [$^{15}$N]Ile1[1-$^{13}$C]Ile3-Ac-I$_3$QGK-NH$_2$ nanoribbons. Rotational sidebands are denoted by asterisks and the down arrow denotes natural abundance $^{13}$C signals. **d** Experimental $^{13}$C{$^{15}$N} REDOR data (black dots) of [$^{15}$N]Ile1[1-$^{13}$C]Ile3-Ac-I$_3$QGK-NH$_2$ nanoribbons. The REDOR curve (black solid line) was fitted by SIMPSON using one intra-strand and two inter-strand $^{13}$C-$^{15}$N spin pairs with corresponding $^{13}$C-$^{15}$N distances of 8.8, 7.4, and 8.1 Å, respectively, which were extracted from the one-residue shift packing mode (inset). The anti-parallel trimer with one-residue shift shown in the inset ($^{13}$C: pink or green sphere and $^{15}$N: blue sphere) was the central core of an Ac-I$_3$QGK-NH$_2$ oligomer of 6 strands × 4 sheets during MD simulations. To demonstrate fitting uncertainty, the dashed lines were calculated using the indicated inter-strand $^{13}$C-$^{15}$N distances

charged head groups of neighboring strands[17,33]. Meanwhile, in order to have the hydrophobic residues between neighboring strands contact, residue shifting is also expected when forming anti-parallel β-sheets[34]. However, more shifting means less inter-strand backbone H-bonding. REDOR results demonstrate anti-parallel β-sheet conformation with one-residue shift for Ac-I$_3$QGK-NH$_2$ self-assembly (Fig. 7a).

The formation of interdigitated bilayers for Ac-I$_3$QGK-NH$_2$, Ac-I$_3$SGK-NH$_2$, and Ac-I$_3$NGK-NH$_2$, and their bilayer thicknesses, which were revealed by SANS and AFM measurements, also confirm the inter-strand packing mode within β-sheets described above. Because Ac-I$_3$SGK-NH$_2$ and Ac-I$_3$NGK-NH$_2$ self-assembled into single bilayer nanoribbons, their thicknesses were accurately determined by SANS to be around 3.3 and 3.2 nm, respectively, in line with the simulated height of ~3.2 nm for the Ac-I$_3$QGK-NH$_2$ oligomer of 6 strands × 4 sheets with one-residue shifting (Supplementary Figure 11). The bilayer thickness could be also deduced from AFM height profiling. This result was slightly larger, such as an AFM height of ~3.8 nm for the Ac-I$_3$SGK-NH$_2$ nanoribbons (Fig. 4b), which may be due to slight conformation changes on drying to the mica surface in addition to measurement errors. However, the same AFM height for both the Ac-I$_3$SGK-NH$_2$ nanoribbons and the single Ac-I$_3$QGK-NH$_2$ bilayers is a result of virtually the same molecular packing mode within β-sheets (Fig. 2c and Fig. 4b). Additionally, although the Ac-I$_3$QGK-NH$_2$ nanoribbons were multilayered, the stacking of bilayered ribbons along the height direction (denoted as the z-direction) was not homogeneous (Fig. 2b, c). Therefore, the

thickness of 8.6 nm derived from SANS was the averaged value and cannot be used to determine the Ac-I$_3$QGK-NH$_2$ bilayer thickness, as in the cases of Ac-I$_3$SGK-NH$_2$ and Ac-I$_3$NGK-NH$_2$. However, such a value suggests that the Ac-I$_3$QGK-NH$_2$ nanoribbons mainly arise from the stacking of two and three bilayers along the z-direction.

Upon forming antiparallel β-sheets, X side-chains are distributed on one side or both sides of a sheet, depending on the degree of residue shifting (Fig. 7a). Since the spacing between neighboring peptide strands in a β-sheet is 4.7 Å, X side-chains lined on each side of an anti-parallel β-sheet are separated by a distance of ~9.4 Å along the sheet (Fig. 7b). The side-chains of Gln and Asn residues have an amide group, containing two H-bonding donors in −NH$_2$ and a H-bonding acceptor in −C=O that can accept one or two H-bonds[35]. The side chain of Ser has a hydroxyl group, capable of donating a proton and accepting one H-bond. Perutz et al. have suggested that continuous stretches of only Gln residues or only Ser residues can act as polar zippers, being capable of linking β-strands into sheets or barrels by H-bonds between their side chains on either side of the sheet, in addition to those between their main chain amides[36,37]. In this case, however, it is unlikely that a polar zipper between X side-chains within a Ac-I$_3$XGK-NH$_2$ (X=Gln, Ser, and Asn) β-sheet will form due to their longer spacing (at least ~9.4 Å and separation by a bulky Ile2 side chain, as shown in Fig. 7b), although the X side chains are only distributed on one side of the antiparallel sheet with a one-residue shift. Because there are two polar Gln, two

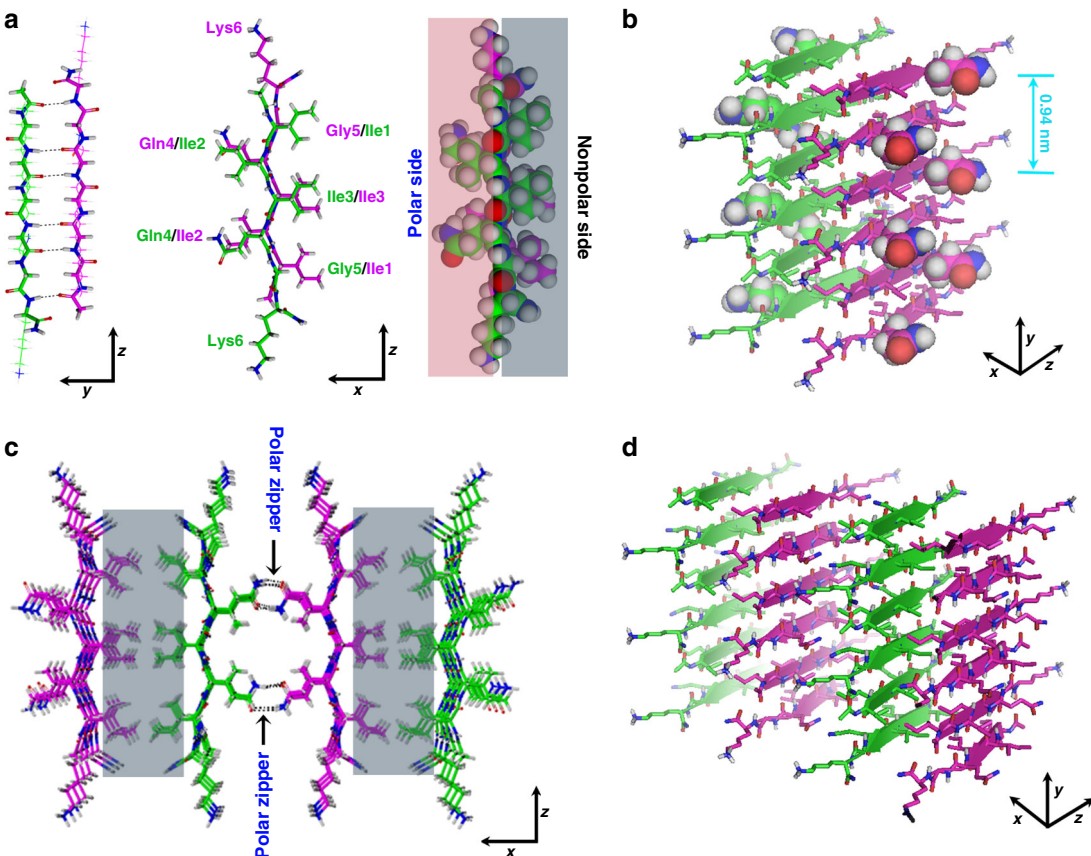

**Fig. 7** Schematic illustration of the self-assembly of Ac-I$_3$XGK-NH$_2$ (X=Q, S, and N) into nanoribbons. Carbon atoms are shown as purple or green, oxygen as red, nitrogen as blue, and hydrogen as gray. **a** Two β-strands pack in an anti-parallel mode with one-residue shifting. The protruding side chains of Ile1, Ile3, and Gly5 constitute the nonpolar face (gray colored) of the β-sheet while the side consisting of the side chains of Ile2, Gln4 (or Ser4 or Asn4), and Lys6 acts as the polar face (pink colored). **b** A pair of β-sheets stabilized by hydrophobic adhesions of two nonpolar faces (back-to-back), in which the backbone of each β-strand is shown as a broad arrow, with protruding X side chains separated by a distance of ~9.4 Å along a sheet. **c** The formation of polar zippers (black arrows) at the polar interface between neighboring β-sheets. The nonpolar interface is colored gray. **c,d** The combination of polar zippers and hydrophobic adhesion intermeshes β-sheets into wide ribbons, as shown from different views

charged Lys, and two hydrophobic Ile residues on this side of a dimer, the side is here termed as the polar surface, in contrast to the other side containing four Ile residues, termed as the nonpolar surface (right panel of Fig. 7a). Upon lateral stacking of Ac-I$_3$XGK-NH$_2$ (X=Gln, Ser, and Asn) β-sheets, their nonpolar sides first tend to associate with one another in a back-to-back mode (Fig. 7b and gray zones in Fig. 7c), similar to the proposed formation of steric zippers at the dry interface in amyloid-like fibrils[30,38]. To overcome the possible electrostatic repulsions from Lys residues between the polar faces and allow β-sheets to undergo further lateral stacking, we propose the formation of polar zippers at the polar interface between neighboring β-sheets (face-to-face) in addition to the hydrophobic contacts of Ile2 residues, as indicated by black arrows in Fig. 7c. Note that in such a zipping mode, the spacing of the X polar side-chains (from two neighboring sheets) along the y-direction is very close, far less than the sheet-sheet spacing of 10.5 Å. This combination of polar Gln/Ser/Asn zippers and hydrophobic adhesive contacts can intermesh β-sheets and significantly overcome their inherent twisting, eventually leading to the formation of wide ribbons, as depicted in Fig. 7c, d. Furthermore, in such a sheet-sheet interaction the hydrophobic contact of Ile side chains is also likely to be maximized, in spite of the one-residue registry shift (Fig. 7c). Additionally, charged side chains of Lys residues tend to protrude from the

assemblies, thus facilitating the loss of curvature in the assemblies and their stability in aqueous solution (Fig. 7c, d and Supplementary Figure 11).

For the polar zippers proposed above, the H-bonding network among X side-chains works as the zipper teeth. The Gln side chain features more H-bonding sites than the Ser one. Therefore, the Gln zipper is stronger than the Ser zipper, and the resulting Ac-I$_3$QGK-NH$_2$ nanoribbons are wider. Asn has the same amide group in the side chain as Gln but the Ac-I$_3$NGK-NH$_2$ nanoribbon was narrower than the Ac-I$_3$QGK-NH$_2$ one, either due to the shorter length of the Asn side chain (4.58 Å for Asn as opposed to 6.11 Å for Gln[39]) or its tendency toward H-bonding with the peptide backbone[40]. However, the formation of polar zippers might require a strict steric confinement among the β-sheets. This is because when the Gln residue is moved along the backbone from the interface to a position between two hydrophobic Ile residues, much thinner ribbons or nanofibers are formed, as demonstrated by Ac-I$_2$QIGK-NH$_2$ (Supplementary Figure 12). The reason why only Ac-I$_3$QGK-NH$_2$ nanoribbons seem to become multilayered (through the stacking of bilayer ribbons) requires further investigation. Although we have previously suggested that His-His (H-H) pairing might be responsible for the formation of flat and multilayered Ac-HI$_4$H-NH$_2$ nanoribbons[41], there are clearly alternative interactions.

The polar zipper mechanism did not occur for the thin and twisted nanofibers formed by Ac-I$_3$GGK-NH$_2$, Ac-I$_3$$^{nor}$VGK-NH$_2$, and Ac-I$_3$LGK-NH$_2$ due to lack of uncharged polar side chains, even although Leu and Ile are well known to form leucine zippers in coiled-coils (preferentially occurring at the positions $d$ and $a$ of the heptad repeat, respectively)[42]. In this case, hydrophobic adhesions between β-sheets primarily drive their lateral stacking. However, these hydrophobic interactions generally provide little specificity and directionality to the stacking of β-sheets, readily leading to their limited lateral stacking and the formation of amyloid-like fibrils[30,38]. The significant sheet twisting within the peptide nanofibers makes the determination of the bilayer thickness difficult.

By rational design of short amphiphilic peptides, we demonstrate the formation of polar zippers between β-sheets rather than between β-strands upon lateral stacking. Such a super-secondary structure is stabilized by H-bonding interactions between polar side chains of neighboring β-sheets. Due to the inherent directionality of H-bonding, the polar zippers can intermesh many β-sheets together into flat and wide ribbons. To the best of our knowledge, this is the first report of the polar zippers formed between β-sheets in peptide self-assembly. By taking advantage of such a super-secondary structural motif, we should be able to design more intricate self-assembled nanostructures in the future, such as peptide barrels.

## Methods

**Peptide synthesis and sample preparation**. The peptides used as well as the isotope-labeled peptide [$^{15}$N]Ile1[1-$^{13}$C]Ile3-Ac-I$_3$QGK-NH$_2$ were all synthesized in our laboratory on a CEM Liberty microwave synthesizer, according to standard Fmoc solid-phase protocols. $^{15}$N-labeled Fmoc-L-Ile-OH and 1-$^{13}$C-labeled Fmoc-L-Ile-OH were purchased from Cambridge Isotope Laboratories, Inc., and other protected amino acids, Rink amide MBHA resin, and other reagents used for peptide synthesis were obtained from GL Biochem Ltd. The peptides were synthesized from their C-termini to N-termini. After cleavage from the resin, the collected peptides were precipitated with cold ethyl ether at least six times and then subjected to further preparative HPLC purification. MALDI-TOF or ESI MS and HPLC characterizations indicated the correct sequences and high purity (> 98%), as shown in Supplementary Figures 13 and 14. The resulting amphiphilic peptides showed high solubility in water and were directly dissolved in Milli-Q water or D$_2$O at a concentration of 8 or 16 mM. After sonication for 30 min, the solution pH or pD values were ~6.5 and then slightly adjusted to 7.0 by using dilute NaOH or NaOD. Note that the amount of NaOH or NaOD added was very limited and did not affect the self-assembled peptide nanostructures.

**AFM and TEM**. AFM measurements were performed on a Bruker MultiMode 8 scanning probe microscope equipped with a NanoScope V controller at room temperature. Samples were adsorbed on a freshly cleaved mica substrate prior to AFM imaging. Height, amplitude, and phase images were concurrently acquired in ScanAsyst mode in air and presented after the first-order flattening. TEM measurements were conducted on a JEOL-1400 electron microscope with an accelerating voltage of 120 kV. TEM samples deposited on a copper grid coated with a carbon support film were stained with 2% uranyl acetate prior to imaging.

**SANS**. SANS experiments were performed on the SANS2D and LOQ diffractometers at the ISIS Pulsed Neutron Source (Rutherford Appleton Laboratory, (Didcot, UK) at 25 °C [21,41] (http://isis.stfc.ac.uk). In brief, the 8 or 16 mM peptide solutions were prepared by directly dissolving peptides in D$_2$O, followed by slightly adjusting the solution pD to 7.0 with dilute NaOD solution. After aging at room temperature for 1 week, they were subjected to SANS measurements at 25 °C. D$_2$O was measured as a matrix background. The 2 or 4 mM peptide solutions were obtained by directly diluting the aged 8 or 16 mM solutions with D$_2$O, and after further aging for 3 h, they were also subjected to SANS measurements at 25 °C. SANS data reduction (to yield $I$ ($q$) vs $q$ on an absolute scale) was performed in the Mantid framework (https://www.mantidproject.org). Model-fitting of the SANS data was performed using the SasView program (version 4.1.0) (http://www.sasview.org).

**CD and FTIR**. CD spectra were recorded on a MOS-450/AC-CD spectrophotometer (Biologic, France) at room temperature, with wavelengths ranging from 260 to 190 nm, and are presented as [$\theta$] (deg cm$^2$ dmol$^{-1}$) versus wavelength. FTIR spectra were collected on a Nicolet 6700 FT-IR equipped with a DGTS detector in absorbance mode. FTIR samples were prepared in D$_2$O and measured in a 0.2 mm pathlength CaF2 solution cell.

**Powder XRD**. XRD measurements were performed on a PANalytical X'pert Pro X-ray diffractometer with Cu Kα radiation ($\lambda = 0.154$ nm) operated at a voltage of 45 kV and a current of 40 mA. After incubation for 1 week, peptide solutions were lyophilized for 2 days to obtain powders for such measurements.

**Solid-state NMR**. NMR experiments were performed on a Varian VNMRS spectrometer operating at 600 MHz for $^1$H, with the 3.2 mm $^1$H/$^{13}$C/$^{15}$N triple-resonance BIO-MAS probe. All spectra were acquired at a MAS speed of 6000 Hz and temperature of 273 K. The $^{13}$C{$^{15}$N} REDOR experiments, where the nucleus in the brackets is the unobserved dephased spin, were performed to detect $^{13}$C-$^{15}$N internuclear distances. Two experiments were conducted alternately for each REDOR mixing time: a control experiment ($S_0$) and a dephasing experiment ($S$) where the $^{15}$N π pulses were turned off and on, respectively. The REDOR decay curves were fitted using the SIMPSON program[43], based on three $^{13}$C-$^{15}$N spin pairs whose $^{13}$C-$^{15}$N distances were extracted from MD simulated conformations.

**MD simulations**. MD simulations were performed using the GROMACS 4.5.5 software package[44]. For all systems, the potential energy of the system was minimized by using the steepest-descent method to relax the initial configurations. Afterward, the peptide molecules were placed in rectangular boxes with periodic boundary conditions and were filled with water molecules. The CharmM 27 all-atom force field and the GBSA implicit-solvent model were used in the simulations[45,46]. The electrostatic interactions were calculated by the Particle Mesh Ewald algorithm[47]. There is no cutoff radius for both the Lennard-Jones interactions and the electrostatic interactions. The temperature of the system was initially set to 10 K and then increased to 203 K. After equilibration for 500 ps, the system was heated to 293 K and kept constant using the Berendsen thermostat for at least 1 ns[48]. The distances of three $^{13}$C-$^{15}$N pairs around one 1-$^{13}$C atom were calculated based on the trimers in the central 4 strands × 2 sheets core of the oligomers of 6 strands × 4 sheets in the MD results.

## Data availability

The authors declare that the data supporting the findings of this study are available within the article and its Supplementary Information file, or are available from the corresponding authors upon request.

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

## Acknowledgements

The work is supported by the National Natural Science Foundation of China under grant numbers 21673293, 21573287, and 21425523. We also thank the UK Engineering and Physical Science Research Council (EPSRC) and Innovate UK for funding support under EP/F062966/1 and KTP008143, and the UK Science and Technology Facilities Council (STFC) for beam time at ISIS. This work benefited from the use of SasView application, originally developed under NSF award DMR-0520547. SasView also contains code developed with funding from the European Union's Horizon 2020 research and innovation programme under the SINE2020 project, grant agreement No 654000.

## Author contributions

J.W., J.R.L., and H.X. conceived the project. M.W., J.W., J.Y., and H.X. designed and performed the experiments. Y.Z., W.Y., and D.W. contributed microscopy. Z.L., X.H., S.M.K., and S.E.R. contributed SANS measurements. P.Z., H.C., and T.A.W. performed MD studies and analyzed conformations. J.D., Y.S., and J.Y. performed NMR measurements. M.W., J.W., J.Y., J.R.L., and H.X. analyzed the data and wrote the manuscript. All authors reviewed the manuscript.

## Additional information

**Competing interests:** The authors declare no competing interests.

