## [Peer Review File · Nature Communications]

Reviewers' comments:

Reviewer #1 (Remarks to the Author):

In this article a number of different peptides have been designed, synthesised and characterised. The resulting morphology has been studied with standard well established techniques, the reasons to the differences of structure observed and hypothesis to explain these differences have been postulated. The SANS data was a very thorough account and excellent fitting to the elliptical cylinder model could be seen. Indeed all of the data was thoroughly analysed and the experiment well designed, with plenty of complementary data.

Some points to consider

Firstly the peptide purity was given to be > 98% by hplc have the authors analysed the peptides with elemental analysis or Amino acid analysis? With the bound counter ions (remaining from the synthesis) I feel that the over all net peptide content would change. As the authors are comparing like for like concentrations I feel that this is an important point to consider.

Table 1- I3QGK data has been presented at 8mM and 8mM – I think this is a typo as it later states it has been diluted 4 fold and should therefore be 2mM and 8mM

With regard to the resulting ribbons – are these kinetically driven or thermodynamically driven? I feel that the existence of these could be a semi stable state and if these peptides were to be studied again over time, would the dominant structural morphology still be ribbons or in the fibrillar state? I feel that the authors need to either present data of the morphology over time or at the very least discuss this point.

The authors have presented the assembly mechanism with the main focus the polar zipper interaction – which is likely to be the key driver in this case. Line 360 stating Q is a stronger H bond than S – which is correct BUT they have overlooked the interaction of the water in the system. With S the hydrophilicity causes better water organisation and would therefore provide an increased energy barrier to its assembly. I feel that the bulk solution conditions should be discussed. They should also state how much NaOH/NaOD was added in each case –you are introducing monovalent cations into solution (and ionic strength of solutions in systems like these have a profound effect).

They mention the CD data and reference the point of 200 being a positive peak (characteristic beta-sheet) – What was the HT voltage at this point? The CD data should not be presented if the HT is >700 which I suspect it is with a bench top instrument.

The discussion and conclusions are based upon the sound reasoning of the observed data. It is of note however, there is no theoretical model or computational model presented to substantiate this. Indeed the computer-generated images are clearly schematics given that the structures have not been energy minimised. They are of value in illustrating the supramolecular structure, however, would the authors be able to run a MD simulation to quantitatively extract the energetic parameters?

Reviewer #2 (Remarks to the Author):

These authors report a very interesting and timely study of self-assembling peptides that form higher order structures. The first order of peptide self-assembly, namely, from individual peptides to form nanofibers, scaffold and hydrogel have been well studied. But the individual peptides that form higher-ordered defined structures remain less well understood. Using a variety of tools

including circular dichroism (CD), FTIR, AFM, TEM, small angle neutron scattering (SANS), these authors studied the higher order of structures.

They systematically designed a series of self-assembling peptides by insertion of 2 amino acids to previously studied peptide Ac-I3XGK-NH₂ (X = Q, S, N, norV, G, and L): a) Ac-I3QGK-NH₂, b) Ac-I3SGK-NH₂, c) Ac-I3GGK-NH₂, d) Ac-I3SGK-NH₂, e) Ac-I3norVG-NH₂, and f) Ac-I3LGK-NH₂. Use the combination and complementary tools, they systematically carried out the detailed studies. Each of these self-assembling peptides behave quite differently with distinct higher order structures. When the X insertion is Q, S or N, these peptides form polar-zipper structures of wide and flat ribbons; but when the X insertion is norV and L, the peptides form amyloid-like structures. These fine-tune studies demonstrate the power of molecular design and control of new materials at single amino acid level.

The experiments for peptide designs are well thought and planned, the complementary methods used for the studies are appropriate, the figures and table in the result section are clear and sound and well explained. The references are properly cited. Their conclusions are justified from their results.

Minor comments:

This reviewer suggests providing more detail information in each of the figure legends and Table 1 so the readers can immediately understand the central ideas presented in each figure and table.

After they make the minor changes, this reviewer recommends publication in Nature Communications.

Reviewer #3 (Remarks to the Author):

The authors this manuscript present an extensive study of designed peptides in which uncharged polar (Q, S and N) and hydrophobic (G, norV and L) residues are inserted in the middle of a small peptide, Ac-I3GK-NH₂. The authors report that the morphology of the assemblies formed by the designed peptides correlates with the type of the inserted residues. Specifically, hydrophobic residues favor the formation of tubular/amyloid-like fibrils and the uncharged polar residues favor ribbon-like assemblies. The authors use a variety of biophysical techniques to draw conclusions about the tertiary structures of the assemblies formed by the designed peptides. Although the observations are quite intriguing, I find that the authors need to address the following issues to make the manuscript suitable for publication in Nature Communication:

1. Most of the conclusions of this manuscript are based on the model, shown in Figure 6 and S8. However, multiple models can be made to fit the low resolution data, which is presented in this manuscript. Alternative methods, such as a solid-state NMR or x-ray diffraction that can show which residues interact to form the assemblies, should be used for validation. Additionally, molecular dynamic simulations should be considered for model building.

2. Although the authors report that the morphology of the peptide assemblies can be modulated by specific residues, it is not clear how the properties of ribbon-like fibrils differ from tubular-like fibrils. What is the potential application of the designed peptides that are described in this manuscript? The authors mention (very briefly) that 2 of the 6 designed peptides (Figure S7 and lines 266-271) template the growth of nanotubes and nanoribbons on silica. However, there is no description in the methods of how these silica based assemblies are made. Most importantly the potential usage of the designed peptides is not quite clear.

3. The authors should show more evidence/data that the fibril assemblies, can be utilized as nanomaterials. What factors modulate the assembly process? Are all monomers recruited into the

fibril? Some of the peptides form heterogeneous mixtures of different species. Thus, how does this sample heterogeneity impact their applications as nanomaterials? Do the assemblies have any unique properties compared to other nanomaterials of peptides?

Overall, unless the authors address the above comments, this manuscript is more suitable for a more specialized journal.

Other comments:

Cross-beta pattern specifically refers to a diffraction pattern recorded from samples of oriented fibrils. Generally cross-beta patterns contain two signature reflections: 4.7Å arcing around the Y-axis and a ~10Å (broad) arcing around X-axis. If the term 'cross-beta' is used, the authors need to show the 2D diffraction pattern that contains the characteristic 4.7Å and 10Å reflections.

The term 'wet interface' is a bit ambiguous. The authors need to clarify it in the text.

What is the error associated with AFM measurements? Specifically how many measurements for the height/length were used for the histograms shown in Figs 1, 3 and 4?

Reply to the Reviewers' Comments

Overall, we greatly appreciate the valuable comments and suggestions of the three reviewers. These have been most helpful in improving the quality of the manuscript and in providing ideas for our future work.

Reviewer #1 (Remarks to the Author):

In this article a number of different peptides have been designed, synthesised and characterised. The resulting morphology has been studied with standard well established techniques, the reasons to the differences of structure observed and hypothesis to explain these differences have been postulated. The SANS data was a very thorough account and excellent fitting to the elliptical cylinder model could be seen. Indeed all of the data was thoroughly analysed and the experiment well designed, with plenty of complementary data.

Reply: We thank Reviewer 1 for recognizing the thorough nature of our work.

Some points to consider

Firstly the peptide purity was given to be > 98% by hplc have the authors analysed the peptides with elemental analysis or Amino acid analysis? With the bound counter ions (reaming from the synthesis) I feel that the overall net peptide content would change. As the authors are comparing like for like concentrations I feel that this is an important point to consider.

Reply: In the revision, we have added the results from MS and HPLC analyses in Supplementary Information (Supplementary Figures 13 and 14), indicating the correct sequences and high purity of the peptides.

On the other hand, we have confirmed a low trifluoroacetic acid (TFA) content in the peptide samples. Because TFA was used to not only cleave peptides from the resin during peptide synthesis but also act as a key component of eluents during peptide purification on preparative reversed phase HPLC column, there is likely some residual TFA in the final peptide products. In addition, as our peptides were synthesized using the standard Fmoc solid-phase synthesis chemistry, such a protocol is unlikely to introduce any other counter ions into the samples. Thus, we determined the concentration of TFA in the prepared peptide solutions by using ion chromatography, in which an IonPac AS11-HC anion-exchange column was applied and 30 mM KOH was used to elute trifluoroacetate. As shown in the following Table R1, there was some residual TFA in the prepared peptide solutions but the contents were very low, typically less than 5%. Thus, such low TFA contents would not affect the peptide concentration significantly. Furthermore, the peptide solutions usually exhibited a pH value of ~6.5 after peptide dissolution,

also suggesting a low TFA content.

Table R1 TFA concentrations (conc.) in peptide solutions determined by ion chromatography

Sample	TFA conc. (ppm)	TFA conc. (mM)	TFA/Peptide (%)
16 mM Ac-I ₃ GGK-NH ₂	18.15	0.162	1.01
8 mM Ac-I ₃ QGK-NH ₂	32.39	0.284	3.55
8 mM Ac-I ₃ SGK-NH ₂	27.22	0.239	2.98
8 mM Ac-I ₃ ^{nor} VGK-NH ₂	44.60	0.391	4.89

Table 1- I3QGK data has been presented at 8mM and 8mM – I think this is a typo as it later states it has been diluted 4 fold and should therefore be 2mM and 8mM.

Reply: In Table 1, Ac-I₃QGK-NH₂ data has been presented at 8 and 2 mM, respectively. At 8 mM, however, a single elliptical cylinder model (ECM) and the combination of two ECM models with different parameters were applied to fit the data, respectively. Correspondingly, two sets of structural parameters were produced for 8 mM Ac-I₃QGK-NH₂ and are given in two columns of Table 1. To avoid confusion, we have introduced three vertical lines in Table 1 of the revision to separate the data belonging to the 3 peptides (Ac-I₃QGK-NH₂, Ac-I₃SGK-NH₂, and Ac-I₃NGK-NH₂).

With regard to the resulting ribbons – are these kinetically driven or thermodynamically driven? I feel that the existence of these could be a semi stable state and if these peptides were to be studied again over time, would the dominant structural morphology still be ribbons or in the fibrillar state? I feel that the authors need to either present data of the morphology over time or at the very least discuss this point.

Reply: The short peptides used in this study are structurally akin to common surfactants, with distinct hydrophilic and hydrophobic segments. Driven by the combination of hydrophobic interactions, hydrogen bonding, and electrostatic repulsions, the peptides can readily undergo self-assembly in aqueous solution to form ordered supramolecular structures. At the same time, the molecular amphiphilicity endows the peptides with high water solubility, as well as their assemblies with high stability in aqueous solution.

The observed peptide ribbons and fibrils were formed typically within one week of incubation, and with further increase of incubation time, there was little variation in their morphologies and dimensions, which could be demonstrated by AFM and SANS characterizations. Furthermore, these peptide assemblies showed high stability in aqueous solution and no precipitates or turbidity

occurred even after several months of storage, in sharp contrast to amyloid fibrils. In the revision, we have added a representative AFM image of Ac-I₃QGK-NH₂ nanoribbons after 3 months of incubation (Supplementary Figure 7 of the revision) and a brief discussion was also given.

The authors have presented the assembly mechanism with the main focus the polar zipper interaction – which is likely to be the key driver in this case. Line 360 stating Q is a stronger H bond than S – which is correct BUT they have overlooked the interaction of the water in the system. With S the hydrophilicity causes better water organisation and would therefore provide an increased energy barrier to its assembly. I feel that the bulk solution conditions should be discussed. They should also state how much NaOH/NaOD was added in each case –you are introducing monovalent cations into solution (and ionic strength of solutions in systems like these have a profound effect).

Reply: Here, the Reviewer raises several concerns, but with the main focus on the effect of the bulk solution conditions.

In fact, we have investigated the effect of pH and ionic strength at the beginning of our study. We found that the pH variation from 3 to 8 and the presence of 5 mM NaCl produced little impact on the self-assembled morphologies. As shown in Figure R1a, Ac-I₃QGK-NH₂ also formed wide and flat ribbons at pH 3.0. In the presence of 5 mM NaCl, Ac-I₃GGK-NH₂ still self-assembled into left-handed twisted nanofibers with structural polymorphism and the narrower ribbons of Ac-I₃SGK-NH₂ still dictated, as shown in Figures R1b and R1c, respectively.

Figure R1 AFM height image of (a) Ac-I₃QGK-NH₂ nanoribbons at pH 3.0, (b) Ac-I₃GGK-NH₂ twisted nanofibers and (c) Ac-I₃SGK-NH₂ nanoribbons at pH 7.0 in the presence of 5 mM NaCl.

As indicated above, the peptide solutions usually exhibited a pH value of ~6.5 after peptide dissolution. For 15 mL of 8 mM Ac-I₃QGK-NH₂ solution that had a pH value of 6.4 after dissolving the peptide product, 9.5 μ L of 1 M NaOH was required to increase the solution pH to 7.0. As a result, the Na⁺ concentration in the solution was only 0.633 mM, far less than 5 mM.

Thus, the amount of NaOH/NaOD added to adjust the solution pH or pD was small and would

not produce profound effect on the self-assembled nanostructures (We have stressed this point in the section of Methods of the revision). At much higher salt concentrations (e.g. 200 mM NaCl), however, it is most likely that the self-assembled morphologies will be changed.

As for the three uncharged polar amino acids G, N, and S, there is not a general consensus about their hydrophobic order, as indicated by the following Table from the paper of Wilce, M. C. J.; et al. (*Anal. Chem.* **1995**, *67*, 1210-1219). However, they seem to have similar hydrophobicity. As a result, we intend not to discuss their effect on water organization and the effect of water organization on assembly, and instead, we have focused on the H-bonding ability of their side chains in the present manuscript.

Table 4. Values of the 12 Previously Published Scales of Amino Acid Hydrophobicity Coefficients*

amino acid	ZIMM	FAUC	BULL	CHOTHIA	JANIN	GUY	HEIJ	HW	KD	MEEK	MEEKR	GUO
Ala	0.83	0.31	-200	0.38	1.70	0.10	-12.04	-0.5	1.8	0.5	1.1	2.2
Arg	0.83	-1.01	-120	0.01	01.0	1.90	39.23	3.0	-4.5	0.8	-0.4	0.9
Asn	0.09	-0.60	80	0.12	0.40	0.48	4.25	0.2	-3.5	0.8	-4.2	-0.8
Asp	0.64	-0.77	-200	0.15	0.40	0.78	23.22	3.0	-3.5	-8.2	-1.6	-2.6
Cys	1.48	1.54	-450	0.50	4.60	-1.42	3.95	-1.0	2.5	-6.8	7.1	2.6
Gln	0.00	-0.22	160	0.07	0.30	0.95	2.16	0.2	-3.5	-4.8	-2.9	-0.2
Glu	0.65	-0.64	-300	0.18	0.30	0.83	16.81	3.0	-3.5	-16.9	0.7	-0.2
Gly	0.10	0.00	0	0.36	1.80	0.33	-7.85	0.0	-0.4	0.0	-0.2	0.0
His	1.10	0.13	-120	0.17	0.80	-0.50	6.28	-0.5	-3.2	-3.5	-0.7	2.2
Ile	2.52	1.80	2260	0.60	3.10	-1.13	-18.32	-1.8	4.5	13.9	8.5	8.3
Leu	3.07	1.70	2460	0.45	2.40	-1.18	-17.79	-1.8	3.8	8.8	11.0	9.0
Lys	1.60	-0.99	-350	0.03	0.05	1.40	9.71	3.0	-3.9	0.1	-1.9	0.0
Met	1.40	1.23	1470	0.40	1.90	-1.59	-8.86	-1.3	1.9	4.8	5.4	6.0
Phe	2.75	1.79	2330	0.50	2.20	-2.12	-21.98	-2.5	2.8	13.2	13.0	9.0
Pro	2.70	0.72	-980	0.18	0.60	0.73	5.82	0.0	-1.6	6.1	4.4	2.2
Ser	0.14	-0.04	-300	0.22	0.80	0.52	-1.54	0.3	-0.8	1.2	-3.2	2.0
Thr	0.54	0.26	-520	0.23	0.70	0.07	-4.15	-0.4	-0.7	2.7	-1.7	0.3
Trp	0.31	2.25	2010	0.27	1.60	-0.51	-16.19	-3.4	-0.9	14.9	17.0	9.5
Tyr	2.97	0.96	2240	0.15	1.50	-0.21	-1.51	-2.3	-1.3	6.1	7.4	4.6
Val	1.79	1.22	1560	0.54	2.90	-1.27	-16.22	-1.5	4.2	2.7	5.9	5.7

* For reference codes, see Table 3.

They mention the CD data and reference the point of 200 being a positive peak (characteristic beta-sheet) – What was the HT voltage at this point? The CD data should not be presented if the HT is >700 which I suspect it is with a bench top instrument.

Reply: In our CD measurements, all the HT voltages have been ascertained not to exceed 600 in the measured wavelength range from 260 to 190 nm, as shown in Figure R2.

Figure R2 HT voltages during the CD measurements of the designed peptides.

The discussion and conclusions are based upon the sound reasoning of the observed data. It is of note however, there is no theoretical model or computational model presented to substantiate this. Indeed the computer-generated images are clearly schematics given that the structures have not been energy minimised. They are of value in illustrating the supramolecular structure, however, would the authors be able to run a MD simulation to quantitatively extract the energetic parameters?

Reply: We have constructed different oligomers of 6 strands \times 4 sheets for MD simulations. The oligomers with anti-parallel conformations were found to be stable during MD simulations (Supplementary Figure 11 of the revision). The structural parameters including the intramolecular and intermolecular atomic distances can be extracted. First, these values helped us determine the isotope-labelled strategy for solid-state NMR experiments. Second, the measured REDOR data measured from Ac-I₃QGK-NH₂ nanoribbons could be well fitted with the simulated values, thereby confirming an anti-parallel β -sheet conformation with one-residue shift for Ac-I₃QGK-NH₂ self-assembly (Figures 5d and Figure 6a of the revision).

In summary, we have carefully taken the comments and suggestions of Reviewer 1 into account during our manuscript revision.

Reviewer #2 (Remarks to the Author):

These authors report a very interesting and timely study of self-assembling peptides that form higher order structures. The first order of peptide self-assembly, namely, from individual peptides to form nanofibers, scaffold and hydrogel have been well studied. But the individual peptides that form higher-ordered defined structures remain less well understood. Using a variety of tools including circular dichroism (CD), FTIR, AFM, TEM, small angle neutron scattering (SANS), these authors studied the higher order of structures.

They systematically designed a series of self-assembling peptides by insertion of 2 amino acids to previously studied peptide Ac-I₃XGK-NH₂ (X = Q, S, N, norV, G, and L): a) Ac-I₃QGK-NH₂, b) Ac-I₃SGK-NH₂, c) Ac-I₃GGK-NH₂, d) Ac-I₃SGK-NH₂, e) Ac-I₃norVG-NH₂, and f) Ac-I₃LGK-NH₂. Use the combination and complementary tools, they systematically carried out the detailed studies. Each of these self-assembling peptides behave quite differently with distinct higher order structures. When the X insertion is Q, S or N, these peptides form polar-zipper structures of wide and flat ribbons; but when the X insertion is norV and L, the peptides form amyloid-like structures. These fine-tune studies demonstrate the power of molecular design and control of new materials at single amino acid level.

The experiments for peptide designs are well thought and planned, the complementary methods used for the studies are appropriate, the figures and table in the result section are clear and sound and well explained. The references are properly cited. Their conclusions are justified from

their results.

Reply: We thank Reviewer 2 for recognizing the thorough nature of our work.

Minor comments:

This reviewer suggests providing more detail information in each of the figure legends and Table 1 so the readers can immediately understand the central ideas presented in each figure and table.

Reply: This is a good suggestion which we have adopted in the revision!

After they make the minor changes, this reviewer recommends publication in Nature Communications.

Reply: We thank Reviewer 2 for recommending publication in Nature Communications.

Reviewer #3 (Remarks to the Author):

The authors this manuscript present an extensive study of designed peptides in which uncharged polar (Q, S and N) and hydrophobic (G, norV and L) residues are inserted in the middle of a small peptide, Ac-I3GK-NH₂. The authors report that the morphology of the assemblies formed by the designed peptides correlates with the type of the inserted residues. Specifically, hydrophobic residues favor the formation of tubular/amyloid-like fibrils and the uncharged polar residues favor ribbon-like assemblies. The authors use a variety of biophysical techniques to draw conclusions about the tertiary structures of the assemblies formed by the designed peptides. Although the observations are quite intriguing, I find that the authors need to address the following issues to make the manuscript suitable for publication in Nature Communication:

1. Most of the conclusions of this manuscript are based on the model, shown in Figure 6 and S8. However, multiple models can be made to fit the low resolution data, which is presented in this manuscript. Alternative methods, such as a solid-state NMR or x-ray diffraction that can show which residues interact to form the assemblies, should be used for validation. Additionally, molecular dynamic simulations should be considered for model building.

Reply: It is unlikely that one could get single crystals of these short amphiphilic peptides for XRD measurements because of their excellent solubility, soft matter nature, and polymorphism even after self-assembly, in sharp contrast to many amyloid peptides. Thus, to address the concern of the reviewer, we have focused our efforts on solid-state NMR and MD simulations.

In solid-state NMR experiments, how to selectively label certain amino acids with ¹³C and ¹⁵N in the peptides is very challenging. In light of MD simulations, we incorporated one 1-¹³C- and one ¹⁵N-labelled Ile amino acid into the sequence of I₃QGK, giving rise to an isotope-labelled peptide [¹⁵N]Ile1[1-¹³C]Ile3-Ac-I₃QGK-NH₂. The measured REDOR data could be exactly described by the simulated distances for the anti-parallel β-sheet packing with one-residue shift,

revealing two nuclear distances of 7.4 and 8.1 Å for the two inter-strand ^{13}C - ^{15}N labeled pairs (Figure 5d of the revision). Furthermore, our MD simulations also indicated two inter-strand ^{13}C - ^{15}N distances of 5.5 and 5.8 Å for the two-residue shift packing mode and of 4.2 and 5.6 Å for the three-residue shift packing mode (Supplementary Figure 9a of the revision). Based on these distances, however, the calculated REDOR curves differ hugely from the experimental data both in shape and S/S_0 values. These combined results from NMR experiments and MD simulations clearly indicate an anti-parallel β -sheet conformation with one-residue shift for Ac-I₃QGK-NH₂ self-assembly.

Accordingly, we have revised the schematic model as shown in Figure 6 of the revision, and the corresponding discussion in the section of Discussion of the revision was also revised substantially.

2. Although the authors report that the morphology of the peptide assemblies can be modulated by specific residues, it is not clear how the properties of ribbon-like fibrils differ from tubular-like fibrils. What is the potential application of the designed peptides that are described in this manuscript? The authors mention (very briefly) that 2 of the 6 designed peptides (Figure S7 and lines 266-271) template the growth of nanotubes and nanoribbons on silica. However, there is no description in the methods of how these silica based assemblies are made. Most importantly the potential usage of the designed peptides is not quite clear.

Reply: Here, the Reviewer raises several concerns, but mainly focusing on the potential applications of the designed peptides.

In fact, we have just begun to explore the possible applications of these peptide assemblies. For example, we have shown that different silica nanostructures were templated from different peptide assemblies in the manuscript. We have added a brief description of how to prepare silica nanostructures from the peptide templates to the Supplementary Information. However, detailed discussion or demonstration of potential applications is beyond the remit of the present manuscript which clearly focuses on the self-assembly process and the formation of polar zippers between β -sheets.

However, we would like to share our opinions on their unique properties and potential applications in the Reply. First, flat nanoribbons have completely different surface curvature from cylindrical nanofibers as well as helical ribbons or nanotubes. As a result, hydrophilic residues, which are mainly located on the assemblies' surface and typically act as functional motifs, have

different local geometries for the three self-assembled architectures. Stupp et al. have demonstrated a high efficiency of interaction between cell receptors and functional RGD motifs on flat peptide nanoribbons, relative to cylindrical peptide nanofibers [Cui, H. et al. *Nano Lett.* **2009**, *9*, 945-951].

Second, uncharged polar amino acids (responsible for the formation of polar zippers in this case) often act as the specific substrates of important enzymes (e.g. transglutaminase, tyrosinase, and kinase), thus endowing the peptide assemblies with potentials in biotechnological applications. In this regard, Ac-I₃QGK-NH₂ has been used for rapid hemostasis [Chen, C.; et al. *ACS Appl. Mater. Interfaces* **2016**, *8*, 17833-17841]. In a study just finished, we realized an extremely quick hemostasis by the mixed use of Ac-I₃QGK-NH₂ and carboxymethyl chitosan (CMCS) (typically less than 10 s). For another example, Ac-I₃YGK-NH₂ that also self-assembles into nanoribbons shows responsiveness to tyrosinase. The latter two examples are unpublished, and we think they deserve to be published in a completely independent paper. We would be very happy to send the data to the editor upon request. We hypothesize that the local curvature affects the dynamic response and efficiency of biochemical and chemical reactions templated by the relevant amino acids. But this aspect requires careful study in future.

3. The authors should show more evidence/data that the fibril assemblies, can be utilized as nanomaterials. What factors modulate the assembly process? Are all monomers recruited into the fibril? Some of the peptides form heterogeneous mixtures of different species. Thus, how this sample heterogeneity impacts their applications as nanomaterials? Do the assemblies have any unique properties compared to other nanomaterials of peptides?

Reply: Here, the Reviewer also raises several interesting aspects. As for the unique properties of the peptide assemblies and their potential applications, please see the above. Furthermore, the peptide assemblies are very stable. We have demonstrated that they could be well kept in aqueous solution for several months, and dilution did not alter their morphologies and dimensions. These characteristics virtually favor their practical biomedical applications.

As for the assembly process, hydrogen bonding, hydrophobic and ionic interactions are the key driving forces. These non-covalent interactions have their respective roles in modulating the self-assembly process. Hydrophobic interactions act as the primary thermodynamic driving force in aqueous solutions, not only propelling peptide monomers to aggregate, but also promoting the

secondary structures formed to undergo further assembly, such as in the formation of coiled-coils and in the lateral stacking of β -sheets. Electrostatic interactions are either repulsive or attractive, depending on the signs of the charges. Charged amino acids are commonly distributed on the surfaces of peptide assemblies and thus dictate their hydration degree and stability in aqueous environments. In this case, the electrostatic repulsions favor the surface curvature formation of assemblies. Hydrogen bonding originates from the peptide backbone or side chains. For self-assembling peptides with β -sheet conformations, backbone hydrogen bonding favors the longitudinal growth of peptide assemblies, resulting in the formation of 1D nanostructures. In the present manuscript, we demonstrate that side-chain hydrogen bonding favors the lateral stacking of β -sheets, in addition to hydrophobic interactions, and acts as a zipper to intermesh them into flat nanoribbons.

Because the designed peptides are structurally similar to surfactants, they behave as the latter to some degree. Upon self-assembly in aqueous solutions, there should be some monomers. However, the monomers are very low in their content and don't affect the conformational homogeneity of peptides in their assemblies, as demonstrated by the ^1H - ^{13}C Cross Polarization (CP) NMR spectrum of $[\text{^{15}N}]\text{Ile1}[1\text{-}^{13}\text{C}]\text{Ile3-Ac-I}_3\text{QGK-NH}_2$ (Figure 5c of the revision). Note that the solid sample for NMR measurements was obtained either by directly lyophilizing the 8 mM solution of $[\text{^{15}N}]\text{Ile1}[1\text{-}^{13}\text{C}]\text{Ile3-Ac-I}_3\text{QGK-NH}_2$ or by ultracentrifugation and there was little difference in their NMR spectra. On the other hand, we have markedly observed fibril polymorphisms for $\text{Ac-I}_3\text{GGK-NH}_2$ self-assembly, similar to many amyloid fibrils. Further studies must examine how such a morphological heterogeneity affects the applications of the nanomaterials.

Overall, unless the authors address the above comments, this manuscript is more suitable for more specialized journal.

Reply: Naturally, we disagree. This was not the conclusion of Reviewers 1 and 2 either. Indeed, this Reviewer seems to be suggesting us to change the focus of the manuscript and to make it more specialized, which is really self-contradicting! In retrospect, we have made major efforts to bring MD and solid-state NMR into this work to consolidate our conclusions, which is to the credit of the Reviewer (we are grateful to the advice), but the multiple questions on various scenarios of application could derail the focus. We reiterate: our manuscript is concerned with the

mechanism behind the directed self-assembly of peptides into controlled nanostructures which are fundamentally interesting (to a wide range of readers in different fields) and which *might* also have interesting applications. However, we are not at the stage to expand the discussion about what those applications are and how we can make use of them.

Other comments:

Cross-beta pattern specifically refers to a diffraction pattern recorded from samples of oriented fibrils. Generally cross-beta patterns contain two signature reflections: 4.7Å arching around the Y-axis and a ~10Å (broad) arching around X-axis. If the term 'cross-beta' is used, the authors need to show the 2D diffraction pattern that contains the characteristic 4.7Å and 10Å reflections.

Reply: We agree with the Reviewer and have deleted the term "cross-β" in the revision.

The term 'wet interface' is a bit ambiguous. The authors need to clarify it in the text.

Reply: The wet interface refers to the interface between β-sheets containing more polar side chains, a term previously coined by Eisenberg et al. in their study of amyloid fibrils (*Nature* **2005**, *435*, 773-777). But in the revision, we have deleted the term in order to avoid confusion.

What is the error associated with AFM measurements? Specifically how many measurements for the height/length were used for the histograms shown in Figs 1, 3 and 4?

Reply: We focus on errors associated with AFM height measurements here. To make the measured heights more reliable from AFM imaging, we have performed height profiling analyses on ~30 individual nanoribbons for each peptide. For Ac-I₃QGK-NH₂, the bilayer thicknesses were found to be mainly between 3.5 nm and 4.0 nm. In the revision, some representative AFM height images and sectional height profiles of Ac-I₃QGK-NH₂ nanoribbons have been given as Supplementary Figure 1.

We thank all the reviewers and the editor for helpful advice and suggestions.

Reviewer #1 (Remarks to the Author):

After reviewing the authors revisions I would like to thank the authors in addressing my comments to a high and a thorough standard. I am satisfied with the response and the amendments to the manuscript.

Reviewer #3 (Remarks to the Author):

The authors of the manuscript provided satisfactory answers to all my critiques. They also carefully addressed the comments of the other reviewers, supporting their answers with experimental data. The added edits improved the quality of the manuscript making it suitable for publication.

POINT-BY-POINT RESPONSE TO THE REVIEWERS' COMMENTS

REVIEWERS' COMMENTS:

Reviewer #1 (Remarks to the Author):

After reviewing the authors revisions I would like to thank the authors in addressing my comments to a high and a thorough standard. I am satisfied with the response and the amendments to the manuscript.

Reply: We thank Reviewer 1 for recognizing our efforts in improving the manuscript quality.

Reviewer #3 (Remarks to the Author):

The authors of the manuscript provided satisfactory answers to all my critiques. They also carefully addressed the comments of the other reviewers, supporting their answers with experimental data. The added edits improved the quality of the manuscript making it suitable for publication.

Reply: We thank Reviewer 3 for recognizing our efforts in improving the manuscript quality and recommending publication in Nature Communications.